# Photo-Generation of Tunable Microwave Carriers at 2 μm Wavelengths Using Double Sideband with Carrier Suppression Modulation

**Di Ji, Zhitao Hu, Nan Ye \*, Fufei Pang and Yingxiong Song**

Key Laboratory of Specialty Optics and Optical Access Networks, School of Communication and Information Engineering, Shanghai University, 99 Shangda Rd., Shanghai 200444, China; fight_d@shu.edu.cn (D.J.); hzt20721358@shu.edu.cn (Z.H.); ffpang@shu.edu.cn (F.P.); herosf@shu.edu.cn (Y.S.)

\* Correspondence: aslanye@shu.edu.cn

**Abstract:** At 2 μm wavelengths (149.9 THz), hollow-core photonics band gap fibers have higher light power damage thresholds, stable polarization states, and lower losses of 0.1 dB/km. Additionally, a thulium-doped fiber amplifier can provide a gain of >35 dB. Specifically, an indium-rich InGaAs photodetector shows a naturally higher photoresponsivity at 2 μm wavelengths than the C-band. Therefore, using tunable photo-generated microwave technology at 2 μm wavelengths could achieve higher photo-to-electric power conversion efficiencies for higher RF output power applications using the same method at the same frequency. Here, a double sideband with the carrier suppression modulation method was experimentally applied on 2 μm wavelengths to generate tunable and stable microwave carriers. Comparison experiments were also applied on the 1.55 μm (193.4 THz)/1.31 μm wavelengths (228.8 THz) based on the same indium-rich InGaAs photodetector. Through normalization on the wavelength-corresponded squared external quantum efficiency to visualize the photo-to-electric power conversion efficiency at different wavelengths under the same input optical signal power, the ratio between the results at 2 μm wavelengths and C/O-band is abstracted as 1.31/1.98, approaching theoretical estimations. This corresponds to a power conversion efficiency increasement of ~1.16 dB/~2.98 dB. To our knowledge, this is the first study on 2 micron wavelengths that proves the corresponding high efficiency power conversion property.

**Keywords:** 2 μm wavelengths; photo-generated microwave carriers; double sideband with carrier suppression modulation





## 1. Introduction

To satisfy the increasing bandwidth demands of 5G/6G wireless networks and RIDAR systems, gigahertz-level high frequency microwave carriers with flexible tuning abilities are required since the generation of high-frequency tunable microwave carriers by traditional electronic methods is limited by the complexity of the system and the large loss of coaxial cables in high RF frequency regions [1–3]. Benefitting from terahertz-level bandwidth and low loss of the fiber, photo-generated microwave carrier technologies can produce high-frequency carriers with tunable properties using simple system settings, which has attracted lots of interests in recent work [4]. Additionally, due to the larger transmission loss of the higher frequency microwave carrier within the air environment, higher output RF power is a focus for satisfying the applications of radar and wireless communications, which requires a higher optical-to-electrical power conversion efficiency for photonics microwave generation systems [5].

Compared with O-band wavelengths, most microwave generation experiments are carried out on C-band wavelengths owing to the advantages of lower fiber loss and more mature devices at the C-band [6]. In this band, photonics microwave generation methods

involve the optical heterodyne method [7], optical injection locking [8], optical phase-lock loop [9], dual-wavelength laser source [10], external modulation method [11], and optoelectronic oscillator (OEO) [12]. Among the abovementioned methods, due to the simple system setup and lower cost, optical microwave carrier generation using the external modulation method has gained more attention in generating tunable, low phase noise, and high-quality microwave carriers despite physical bottlenecks existing in the standard single-mode fibers (SSMFs), involving random polarization state property, damage, and nonlinear effects from the high input power [13,14]. Moreover, the erbium-doped fiber amplifier (EDFA) working around 1.55 μm wavelengths struggles to provide a >30 dB optical gain for generating high-power microwave carriers [15]. Those challenges limit improvements in photo-generated microwave systems. Therefore, it is necessary to explore new and essential settlements that depend on novel optical fiber technologies.

Around 2 μm wavelengths, benefitting from the special micro-structure of optical fiber, hollow-core photonics band gap fibers (HC-PBGFs) not only have near-vacuum delays [16] and lower nonlinearities [17] but also have higher light power damage thresholds [18], stable polarization states [19], and lower losses of 0.1 dB/km [20]. Additionally, high-power photo-generated microwave carriers can be obtained due to the gain of >35 dB offered by the thulium-doped fiber amplifier (TDFA) working at 2 μm wavelengths [21]. Specifically, assuming the same external quantum efficiency, a naturally higher photoresponsivity at 2 μm wavelengths than that of the 1.55 μm wavelengths was experimentally observed from the same indium-rich InGaAs photodiode with cut-off wavelengths of 2 μm [22]. Therefore, using photonics microwave generation technology at 2 μm wavelengths can take advantage of the higher opto-electric energy conversion efficiency for higher output RF power applications based on the same method and at the same RF frequency.

In this paper, 7–12 GHz, stable, frequency-doubling, and tunable microwave carriers were experimentally obtained at 2 μm using the double sideband carrier suppression (DSB-CS) modulation method for a comparison with the results at 1.55 μm/1.31 μm wavelengths tested using the same method at the same frequency. In the experiment, the light side band power (~−7.7 dBm) output from the Mach–Zehnder Modulators (MZMs) was realized at those three different regions. Upon using the same indium-rich InGaAs PIN photodetector (cut-off wavelength covering 1.31 μm to 2 μm wavelengths), a higher microwave carrier power per squared external quantum efficiency can be obtained at the 2 μm wavelengths than at the 1.55 μm/1.31 μm wavelengths, which approaches the theoretical estimations. This confirms the large opto-electric energy conversion efficiency obtained at the 2 μm wavelengths, which may also be the first time it is proven, according to our knowledge. Compared with our previously published conference proceeding [22], here, a formulaic derivation of the power generation efficiency for different wavelengths using the same method and the newest experimental results obtained at the O-band are added.

## 2. Basic Theory

### 2.1. The Double Sideband with Carrier Suppression (DSB-CS) Modulation Method

Microwave carriers can be generated by means of DSB-CS modulation method with the advantages of input wavelength independence and microwave frequency tunability/doubling. In addition, the optical signal power generated at different wavelength regions after the MZMs can be calibrated by adjusting the input light power and the DC bias of the modulators. This provides convenience in comparing the photonics-generated microwave power at different wavelengths based on the same photodiode. Here, it is assumed that a single drive Mach–Zehnder Modulator (MZM) was used to generate the microwave carriers. Then, the input optical carrier electric field of the MZM is expressed as Equation (1).

$$E_{in}(t) = E_0 exp(j\omega_c t) \tag{1}$$

where $E_{in}(t)$ is the input optical carrier electric field, $E_0$ represents the amplitude of the optical carrier electric field, and $\omega_c$ represents the angular frequency of the input optical carrier.

Being loaded on the single-driving MZM, the RF modulation signal is expressed in Equation (2).

$$V_{RF}(t) = V_{RF}\sin(\omega_{RF}t + \varphi) \tag{2}$$

where $V_{RF}(t)$ represents the RF modulation signal, $V_{RF}$ represents the voltage of the RF modulation signal, $\omega_{RF}$ represents the angular frequency of the RF modulation signal, and $\varphi$ is the phase of the sinusoidal modulated electrical signals generated by the signal generator.

When the optical separation ratio of the Y-branches of the MZM is set as 0.5/0.5, after outputting the combined Y-branch of the MZM, the optical signal $E_{MZMout}(t)$ is expressed by Equation (3), which is expanded based on the first kind of Bessel function.

$$
\begin{aligned}
E_{MZMout}(t) \\
= \frac{E_0}{2}\left\{\sum_{-\infty}^{+\infty} J_n\left(\frac{\pi V_{RF}}{2V_\pi}\right)exp(j\omega_c t + jn\omega_{RF}t + jn\varphi)\left[exp\left(j\frac{\pi V_{DC}}{2V_\pi}\right) + (-1)^n exp\left(-j\frac{\pi V_{DC}}{2V_\pi}\right)\right]\right\} \\
= \frac{E_0}{2}\left\{\sum_{-\infty}^{+\infty} J_n(m)exp(j\omega_c t + jn\omega_{RF}t + jn\varphi)\left[exp(j\Phi_{DC}) + (-1)^n exp(-j\Phi_{DC})\right]\right\}
\end{aligned}
\tag{3}
$$

where $V_\pi$ represents the half-wave voltage of the MZM. $V_{DC}$ is the DC bias voltage of the MZM. $m = (\pi V_{RF})/(2V_\pi)$ is the modulation index of the MZM. $\Phi_{DC} = (\pi V_{DC})/(2V_\pi)$ is the optical phase changes caused by the DC bias voltage of the MZM, and $\Delta\Phi_{DC} = (\pi V_{DC})/V_\pi$ is the optical phase difference between the upper and lower channels of the Y-branch caused by the DC bias voltage of the MZM.

When the DSB-CS modulation method is applied, the value of $\Delta\Phi_{DC}$ is maintained as $\pi$ by adjusting the DC bias voltage of the MZM to $V_\pi$. At this moment, the power of the optical carrier is kept at a minimum as zero. The electric field of the high-order optical sidebands can be ignored because of the properties of the Bessel function. Then, Equation (3) is simplified to Equation (4) based on Euler's Formula.

$$
\begin{aligned}
E_{MZMout}(t) &= \frac{E_0}{2}\left\{\sum_{-\infty}^{+\infty} J_n(m)exp(j\omega_c t + jn\omega_{RF}t + jn\varphi)\cdot\left[j - (-1)^n j\right]\right\} \\
&= E_0 J_1(m)exp\left(j\omega_c t + j\omega_{RF}t + j\varphi + j\frac{\pi}{2}\right) + E_0 J_{-1}(m)exp\left(j\omega_c t - j\omega_{RF}t - j\varphi + j\frac{\pi}{2}\right)
\end{aligned}
\tag{4}
$$

As shown in Equation (4), the DSB-CS modulation is realized with the suppressed optical carrier power ($n = 0$) demonstrating an optical carrier wavelength independent property. Additionally, the first-order optical sidebands ($n = \pm1$) with the paired angular frequency of ($\omega_c + \omega_{RF}$) and ($\omega_c - \omega_{RF}$) are maintained.

Then, the optical signal power $P_{MZMout}(t)$ illuminated on photodetector can be expressed in Equation (5).

$$
\begin{aligned}
P_{MZMout}(t) &\propto \lim_{T\to\infty}\int_{-T}^{T}\frac{1}{2T}\left|E_{MZMout}(t)E_{MZMout}^*(t)\right|dt \\
&\propto \begin{bmatrix} E_0^2 J_1^2(m) + E_0^2 J_{-1}^2(m) + 2E_0^2 J_1(m)J_{-1}(m)\cos(j2\omega_{RF}t + j2\varphi) + \\ 2E_0^2 J_1(m)J_{-1}(m)\cos(j2\omega_c t + j\pi) \end{bmatrix}
\end{aligned}
\tag{5}
$$

where the first two items, $E_0^2 J_1^2(m)$ and $E_0^2 J_{-1}^2(m)$, are the spectral response terms that represent the DC components. The last two terms are the time-varying response terms of optical power, which are related to the frequency response of the photodetector.

After collection and beating at the receiver side, the microwave carrier with the angular frequency of ($2\omega_{RF}$) is generated. Obviously, by changing the frequency ($\omega_{RF}$) of the RF signal, which can be achieved when driving the MZM at the transmitting side, a tunable microwave carrier with doubling-frequency ($2\omega_{RF}$) can be successfully achieved at the receiver side. Specifically, through the adjustment of the driving voltage ($V_{RF}$) corresponding to $J_1(m)J_{-1}(m)$ or the input light power corresponding to $E_0^2$, the same $P_{MZMout}(t)$ can be realized regardless of the differences between MZMs working in different wavelength regions, indicating the wavelength-transparent property.

Therefore, if the wavelength dependance of the detector is not considered, the microwave carrier generated by the DSB-CS modulation method would demonstrate prop-

erties involving independence from the optical carrier wavelength, frequency doubling, and tunability.

### 2.2. RF Power Generation Efficiency Based on the Indium-Rich PIN Photodiode Using the DSB-CS Modulation Method

Due to the optical carrier wavelength-independent property when applying the DSB-CS modulation method, microwave carrier power generation with the optical signal illuminated on one single PIN photodiode would only be related to the detector photoresponsivity, which is proportional to the input light wavelength.

When the light is illuminated on the PIN photodiode for photo-electric energy conversion, the value of the photo-generated current ($I_{pin}$) generated from the photodiode is expressed as Equation (6).

$$I_{pin} = RP_{MZMout} \tag{6}$$

where $R$ represents the photoresponsivity of the photodiode.

The photoresponsivity ($R$) of the photodiode is expressed in Equation (7).

$$R = \frac{\eta e \lambda}{hc} \tag{7}$$

where $\eta$ is the external quantum efficiency of the photodiode. $\lambda$ denotes the wavelength of the optical carrier. $e = 1.6 \times 10^{-19}$ C, $h = 6.625 \times 10^{-34}$ J · s, and $c = 3 \times 10^8$ m/s. It can be seen that the photoresponsivity is proportional to the wavelength of the input light, demonstrating a higher photoresponsivity at longer wavelengths.

Ignoring the DC components and the AC component that exceed the cut-off response frequency of the photodetector, Equation (6) can be expressed as Equation (8) based on Equations (5) and (7).

$$I_{pin} \propto \left( \frac{\eta e \lambda}{hc} \right) \cdot 2E_0^2 J_1(m) J_{-1}(m) \cos(j2\omega_{RF}t + j2\varphi) \tag{8}$$

Therefore, the microwave carrier power ($P_{E\lambda}$) converted from the input light signal through the detector and measured by the electric spectrum analyzer can be expressed as Equation (9).

$$P_{E\lambda} \propto \left( I_{pin} \right)^2 \cdot Z_L \propto R^2 \cdot P_{MZMout}^2 \cdot Z_L \propto \left( \frac{\eta e \lambda}{hc} \right)^2 \cdot P_{MZMout}^2 \cdot Z_L \tag{9}$$

where $Z_L$ is the input impedance of the electric spectrum analyzer, which is generally a constant.

Referring to different input wavelengths such as $\lambda_1$ and $\lambda_2$, Equation (10) shows the ratio between the power of the microwave carriers generated after the photodetector at wavelengths $\lambda_1$ and $\lambda_2$ based on Equation (9).

$$\frac{P_{E\lambda_1}}{P_{E\lambda_2}} = \frac{R_{\lambda_1}^2 \cdot P_{\lambda_1 MZMout}^2 \cdot Z_L}{R_{\lambda_2}^2 \cdot P_{\lambda_2 MZMout}^2 \cdot Z_L} = \frac{(\eta_{\lambda_1}\lambda_1)^2 \cdot P_{\lambda_1 MZMout}^2}{(\eta_{\lambda_2}\lambda_2)^2 \cdot P_{\lambda_2 MZMout}^2} \tag{10}$$

Here, the values of $P_{\lambda_1 \, MZMout}$ and $P_{\lambda_2 \, MZMout}$ are set as the same for comparing the electrical power generation efficiencies at different wavelengths based on the same photodiode to demonstrate the relationship between the photo-generated microwave carrier power and the incident light wavelength of the photodiode. Then, Equation (10) can be expressed as Equation (11).

$$\frac{P_{E\lambda_1}}{P_{E\lambda_2}} = \frac{(\eta_{\lambda_1}\lambda_1)^2 \cdot P_{\lambda_1 MZMout}^2}{(\eta_{\lambda_2}\lambda_2)^2 \cdot P_{\lambda_2 MZMout}^2} = \frac{(\eta_{\lambda_1}\lambda_1)^2}{(\eta_{\lambda_2}\lambda_2)^2} \tag{11}$$

Under the ideal circumstance, the external quantum efficiency $\left(\eta_{\lambda_1}\right)$ and $\left(\eta_{\lambda_2}\right)$ are the same. Therefore, the ratio is shown as Equation (12).

$$\frac{P_{E\lambda_1}}{P_{E\lambda_2}} = \frac{\lambda_1^2}{\lambda_2^2} \tag{12}$$

It is shown in Equation (12) that, if $\lambda_1$ is longer than the $\lambda_2$, $P_{E\lambda_1}$ is larger than $P_{E\lambda_2}$ demonstrating a higher microwave carrier power obtained at the longer wavelengths.

Actually, for a real detector, $\eta_{\lambda_1} \neq \eta_{\lambda_2}$. To exclude the affection of such a case for obtaining the opto-electric power conversion efficiency, the microwave carrier power is normalized to the square of the referring external quantum efficiency. Therefore, we have the function in Equation (13).

$$\left(\frac{P_{E\lambda_1}}{\eta_{\lambda_1}^2}\right) \bigg/ \left(\frac{P_{E\lambda_2}}{\eta_{\lambda_2}^2}\right) = \frac{\lambda_1^2}{\lambda_2^2} = r_{n(\lambda_1/\lambda_2)} \tag{13}$$

where $\left(P_{E\lambda_{1\ or\ 2}}\right)/\left(\eta_{\lambda_{1\ or\ 2}}^2\right)$ represents the microwave carrier power per squared external quantum efficiency referring to the opto-electric power conversion efficiency under the same input optical signal power. $r_{n(\lambda_1/\lambda_2)}$ represents the ratio between the normalized microwave carrier power at the wavelengths of $\lambda_1$ and $\lambda_2$, indicating the variation of the opto-electric power conversion efficiency due to the difference of the input wavelengths.

It is obviously shown from Equation (13) that $r_{n(\lambda_1/\lambda_2)} > 1$ when $\lambda_1 > \lambda_2$. This indicates a higher microwave carrier power per squared external quantum efficiency corresponding to a larger opto-electric power conversion efficiency obtained at the longer wavelengths.

Based on the deduction of the abovementioned equations, it is theoretically demonstrated that a higher photo-to-electric power conversion efficiency can be achieved at longer wavelengths under the same optical signal power benefitting from the higher photoresponsivity of the detector at longer wavelengths.

## 3. Experimental Setup

At the transmitter side, the optical sources were provided by single mode laser diodes (LDs, EP2000-DM-B at 2 µm wavelengths by Eblana Photonics at Dublin, Ireland/NBD1550-001S8 at C-band by Netopto at Shenzhen, China/TSL-550 at O-band by Santec Corporation at Aichi-ken, Japan). The working temperature and the current of the LD were maintained as constants by employing a Laser Diode Controller (LDC, Newport LDC-3744C by Newport Corporation at Montana, America). Because of the starting power threshold of the optical amplifiers and the large loss in MZMs, the optical carrier outputted from the LD was amplified by a commercial optical amplifier (TDFA, CTFA-PB-SM-33-BW3 at 2 µm wavelengths by Keopsys at Lannion, France/EDFA, LTRAN LOA3000 at C-band by Luster at Beijing, China/SOA, AEON006 at O-band by Aeon Laser at Florida, America) before entering the MZMs. The optical amplifier was protected by the isolator (OP-112110333 at 2 µm wavelengths by Idealphotonics at Shenzhen, China/HOPECOM 150128001 at C-band by Hopecom Optic Communications at Shanghai, China/HOPECOM 160706004 at O-band by Hopecom Optic Communications at Shanghai, China) from the reflection power. The best modulation efficiency of the polarization-sensitive MZMs was achieved by adopting a polarization controller (PC) to adjust the polarization state. The RF modulation signals (sinusoidal type) were applied by a signal generator (1465F-V by Ceyear at Qingdao, China) to drive the Mach-Zehnder Modulators (MX2000-LN-10 at 2 µm wavelengths by iXBlue Photonics at Besancon, France/FTM7938EZ at C-band by Fujitsu Optical Components Limited at Kanagawa-ken, Japan/MX1300-LN at O-band by iXBlue Photonics at Besancon, France), and the phase noise was as low as—120 dBc/Hz at 100 kHz offset (3 GHz $\leq f \leq$ 20 GHz). At the receiving end, the photo-to-electric power conversion was achieved by employing an indium-rich InGaAs diode (ET-5000F by Electro-Optics Technology at Michigan, America) with the cut-off wavelength covering 1.31 µm to 2 µm

wavelengths. Optical spectrums of the optical signals after the MZMs and electric spectrums of the microwave carriers generated were observed by applying an optical spectrum analyzer (OSA, AQ6375 by Yokogawa Electric Corporation at Tokyo, Japan) and electric spectrum analyzer (ESA, N9030A by Keysight Technologies at California, America). In the experiment, the C-band single mode fibers were used to connect the optical devices that were also able to work at the 2 μm/1.31 μm wavelengths, and the experimental setups at the 2 μm/1.55 μm/1.31 μm wavelengths were similar. Figure 1 shows the experimental setup for microwave carrier generation at the 2 μm/1.55 μm/1.31 μm wavelengths using the DSB-CS modulation method.

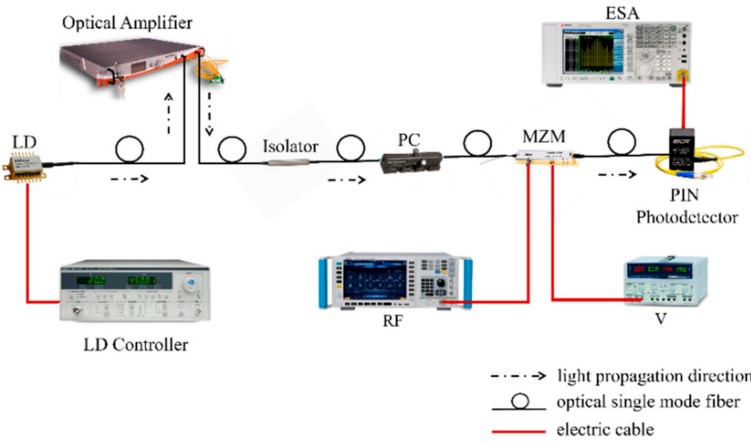

**Figure 1.** Experimental setup of the photonics-generated microwave carrier system at 2 μm/1.55 μm/1.31 μm wavelengths using the DSB-CS modulation method.

## 4. Results and Discussions

Adopting the experimental setup shown in Figure 1, the experimental results at the 2 μm/1.55 μm/1.31 μm wavelengths were acquired based on the same indium-rich InGaAs photodetector (cut-off wavelength covering 1.31 μm to 2 μm wavelengths), similar experimental setups consisting of the same 18 dBm sinusoidal-type RF signal of the MZMs, and similar optical spectrums output from the MZMs with ~−7.7 dBm optical peak power.

At the spectral resolution (~0.05 nm), the optical spectrums output by the 2 μm/1.55 μm/1.31 μm wavelength MZMs working in the DSB-CS modulation method were obtained in order to generate the 12 GHz microwave carriers, as shown in Figure 2a,b. Obviously, the optical spectrums demonstrated similar optical signals using the same DSB-CS modulation method and practically the same optical peak power (~−7.7 dBm) at those three different wavelengths. An Optical Side Mode Suppression Ratio (OSMSR) up to 49 dB (46.3 dB at 1.55 μm; 49.3 dB at 2 μm) and suppression of the higher mode components in the Bessel expansion ($J_2(0.29) = 0.01$, $J_3(0.29) = 0.0005$ when the MZM modulation index $m = (\pi V_{RF})/(2V_\pi) = 0.29$) suppresses the effect of the higher-order sidebands away from the target bands and kinks on the shoulders of the modulated peaks.

Figure 3a–c have shown the electric spectrums of the 12 GHz frequency-doubling photo-generated microwave carriers using the MZMs at the DSB-CS modulation method at the 2 μm/1.55 μm/1.31 μm wavelengths. It is shown that the high-quality microwave carriers with low phase noise were generated and that the Side Mode Suppression Ratios (SMSRs) were 23.2 dB at 2 μm wavelengths, 23.6 dB at 1.55 μm wavelengths, and 22.1 dB at 1.31 μm wavelengths. Therefore, only the peak power generated from the first-order side band was considered during the following measurements or discussion due to the high SMSR value.

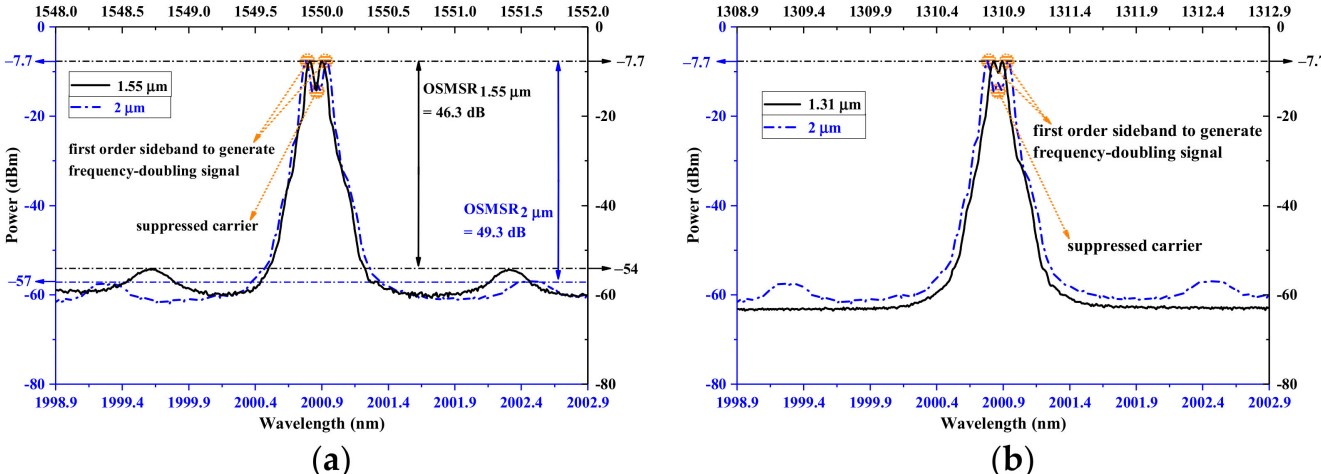

**Figure 2.** The optical spectrums after the DSB-CS modulation using the (**a**) 2 μm MZM (at 2000.86 nm)/1.55 μm MZM (at 1549.96 nm); (**b**) 2 μm MZM (at 2000.86 nm)/1.31 μm MZM (at 1310.82 nm).

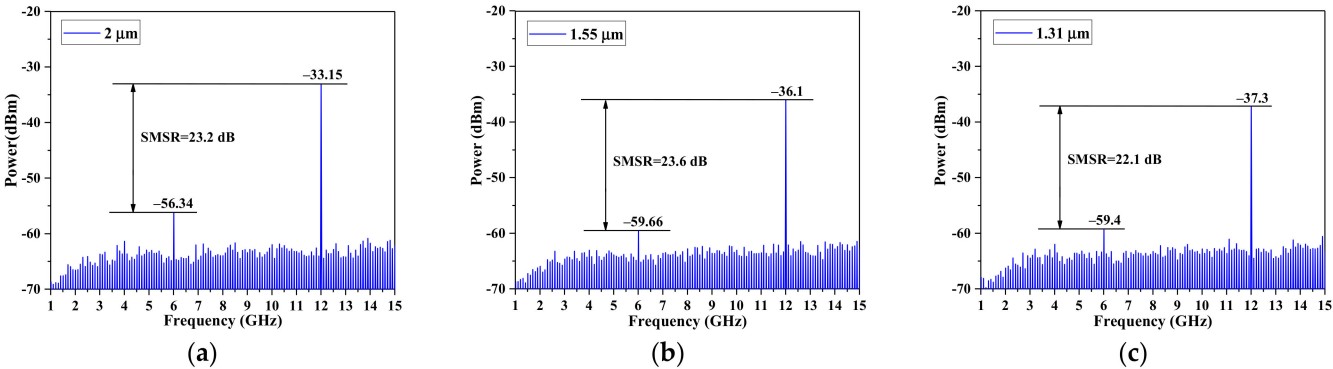

**Figure 3.** The electric spectrums for the 12 GHz microwave carriers based on photonics generation using the DSB-CS modulation method at (**a**) 2 μm, (**b**) 1.55 μm, and (**c**) 1.31 μm.

Figure 4a–c give the tunable frequency-doubling microwave carriers obtained at the same setting parameters as the 12 GHz high-quality microwave carriers generated and shown in Figure 3a–c. The 7–12 GHz, stable, frequency-doubling, and tunable microwave carriers were generated at those three different wavelengths because of the 12 GHz bandwidth limitation for the MZMs and the diode, as shown in Figure 4a–c. As shown in Table 1, the phase noise of about −100 dBc/Hz at 10 MHz and linear dynamic range up to 20 dB were obtained at the tested frequency of 7–12 GHz for the 2 μm wavelengths.

In addition, the phase noise of our setup obtained at 2 μm wavelengths is still much higher than the performance of the available commercial electronic microwave synthesizers at the same frequency of 10 GHz (see Table 1), which suffers from noise issues such as ASE noise.

To highlight the influence of the wavelength $\lambda$ on the photo-to-electric power conversion efficiency, the generated microwave carrier power is normalized to the square of the corresponding external quantum efficiency ($\eta$) under the same optical signal power used as the input. Then, based on Equation (13), the theoretical ideal power conversion efficiency variation between 2 μm and 1.55 μm wavelengths is $r_{ni(2\ \mu m/1.55\ \mu m)} = \left(\lambda^2_{2\ \mu m}\right)/\left(\lambda^2_{1.55\ \mu m}\right) = 1.66$. Experimentally, the generated microwave carrier power of $P_{E2\ \mu m}$ and $P_{E1.55\ \mu m}$ can be obtained at the 2 μm and 1.55 μm wavelengths using the abovementioned setup under the same optical signal peak power of −7.7 dBm. Therefore, the measured power conversion efficiency variation $r_{nm(2\ \mu m/1.55\ \mu m)} = \left(P_{E2\ \mu m}/\eta^2_{2\ \mu m}\right)/\left(P_{E1.55\ \mu m}/\eta^2_{1.55\ \mu m}\right)$

can be sorted out as in Figure 5a considering the external quantum efficiency ratio of $(\eta_{2\ \mu m}/\eta_{1.55\ \mu m}) = 1.227$.

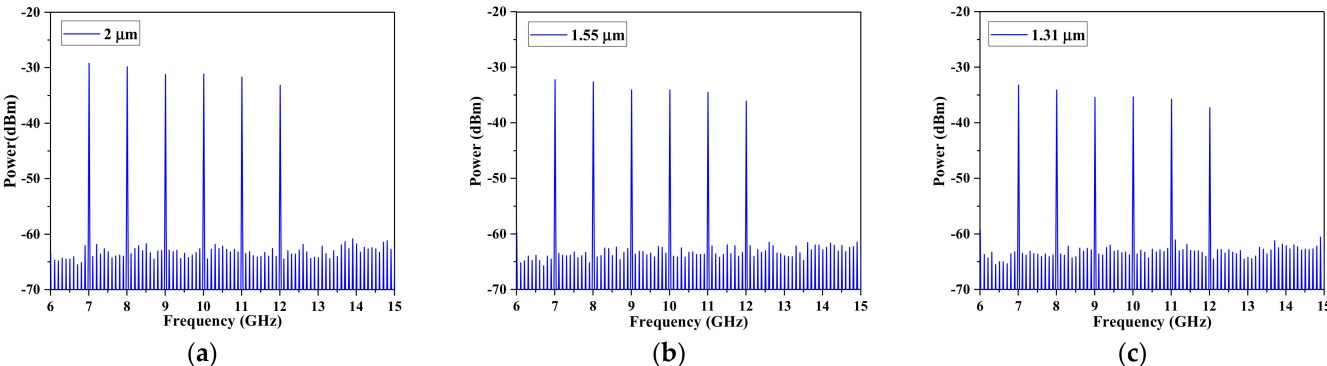

**Figure 4.** The generated 7–12 GHz tunable frequency-doubling microwave carriers after the DSB-CS modulation using the optical carrier at (**a**) 2 μm, (**b**) 1.55 μm, and (**c**) 1.31 μm.

**Table 1.** The phase noise and linear dynamic range of the 7–12 GHz microwave carriers at 2 μm wavelengths.

| Frequency (GHz) | Phase Noise at 2 μm (dBc/Hz at 10 MHz) | Phase Noise (dBc/Hz at 10 MHz, KRATOS-SF6219) | Phase Noise (dBc/Hz at 10 MHz, Micro Lambda Wireless MLVS-0520) | Phase Noise (dBc/Hz at 10 MHz, AnaPico APSYN140) | Linear Dynamic Range at 2 μm (dB, RF Power: −4.5~18 dBm) |
|---|---|---|---|---|---|
| 7 | −112 | | | | 22.5 |
| 8 | −109 | | | | 21.9 |
| 9 | −108 | | | | 21.4 |
| 10 | −107 | −110 | −138 | −136 | 21.7 |
| 11 | −107 | | | | 21.4 |
| 12 | −105 | | | | 20.1 |

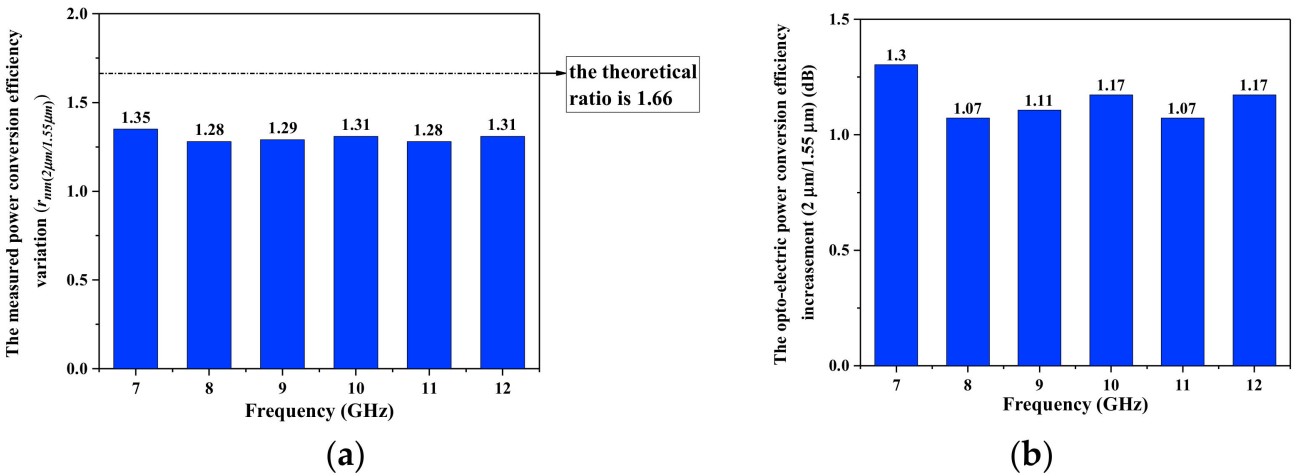

**Figure 5.** (**a**) The theoretical/measured power conversion efficiency variation $r_{nm(2\ \mu m/1.55\ \mu m)} = (P_{E2\ \mu m}/\eta_{2\ \mu m}^2)/(P_{E1.55\ \mu m}/\eta_{1.55\ \mu m}^2)$; (**b**) the opto-electric power conversion efficiency increasement (2 μm/1.55 μm).

As shown, the $r_{\mathrm{nm(2~\mu m/1.55~\mu m)}}$ values measured are close to the theoretical estimation (see Figure 5a). Correspondingly, an opto-electric power conversion efficiency increasement of $10log_{10}\left[(P_{E2~\mu m}/\eta_{2~\mu m}^2)/(P_{E1.55~\mu m}/\eta_{1.55~\mu m}^2)\right]$ can be obtained, as seen in Figure 5b, with the average value of ~1.16 dB.

Similarly, based on Equation (13), the ideal power conversion efficiency variation between 2 μm and 1.31 μm wavelengths is $r_{ni(2~\mu m/1.31\mu m)} = \left(\lambda_{2~\mu m}^2\right)/\left(\lambda_{1.31~\mu m}^2\right) = 2.33$. Experimentally, the generated microwave carrier power of $P_{E2~\mu m}$ and $P_{E1.31~\mu m}$ can be obtained at 2 μm and 1.31 μm wavelengths by using the setup mentioned above under the same optical signal peak power of −7.7 dBm. So that, the measured power conversion efficiency variation $r_{\mathrm{nm(2~\mu m/1.31\mu m)}} = (P_{E2~\mu m}/\eta_{2~\mu m}^2)/(P_{E1.31~\mu m}/\eta_{1.31~\mu m}^2)$ can be sorted out as Figure 6a considering the external quantum efficiency ratio of $(\eta_{2~\mu m}/\eta_{1.31~\mu m}) = 1.146$.

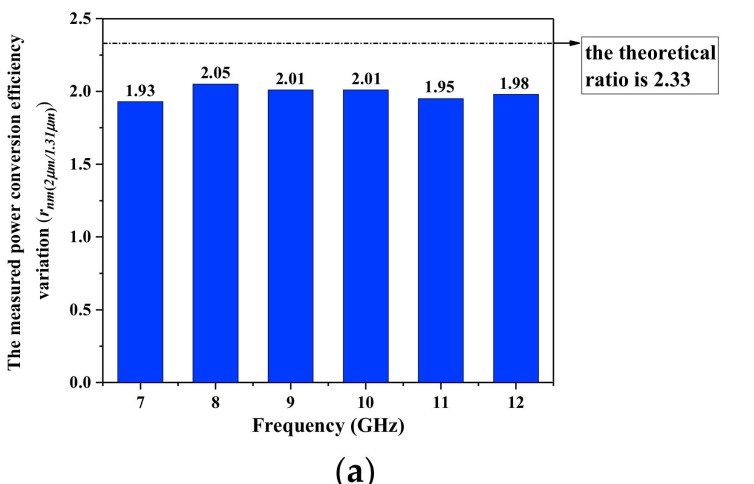 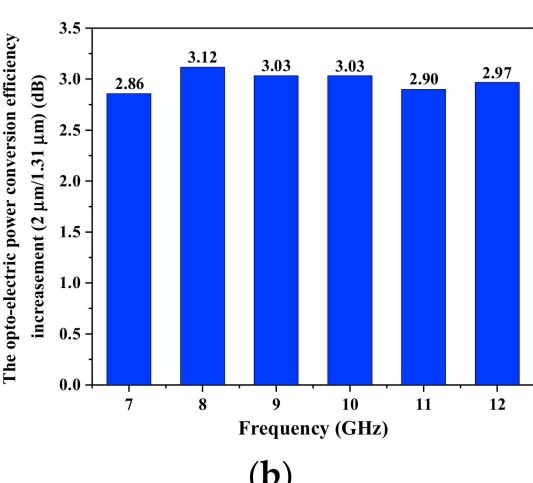

(a)   (b)

**Figure 6.** (**a**) The theoretical/measured power conversion efficiency variation $r_{\mathrm{nm(2~\mu m/1.31~\mu m)}} = (P_{E2~\mu m}/\eta_{2~\mu m}^2)/(P_{E1.31~\mu m}/\eta_{1.31~\mu m}^2)$; (**b**) the opto-electric power conversion efficiency increasement (2 μm/1.31 μm).

It is shown that all of the measured values are close to the theoretical estimation (see Figure 6a) and that the opto-electric power conversion efficiency increasement of $10log_{10}\left[(P_{E2~\mu m}/\eta_{2~\mu m}^2)/(P_{E1.31~\mu m}/\eta_{1.31~\mu m}^2)\right]$ can be correspondingly obtained (see Figure 6b) with the average value of ~2.98 dB.

From Figures 5 and 6, it can be seen that a higher opto-electrical power conversion efficiency was obtained at the 2 μm wavelengths compared with at the 1.55 μm and 1.31 μm wavelengths under the same optical signal power. This aligns with the theoretical analysis of the larger photoresponsivity that can be obtained at longer input wavelengths $(R = (\eta e\lambda)/(hc))$.

## 5. Conclusions

In summary, 7–12 GHz, stable, frequency-doubling, and tunable microwave carriers have been theoretically and experimentally demonstrated at 2 μm wavelengths by applying the DSB-CS modulation method. A similar experimental setup has also been built for the 1.55 μm/1.31 μm wavelengths region consisting of the RF modulation signal power (18 dBm), light side band power (~−7.7 dBm), and spectral resolution (0.05 nm). The same indium-rich InGaAs photodetector covering the working band until 2 μm wavelengths was used. The microwave carrier power was normalized to the wavelength-corresponded squared external quantum efficiency, and the ratio of generated microwave carrier power between the results at 2 μm and the C/O-band is obtained as 1.31/1.98, demonstrating a power conversion efficiency increasement of ~1.16 dB/~2.98 dB due to the higher detector photoresponsivity at longer wavelengths, which aligns with the theory. That proves the

higher power conversion efficiency at 2 micron wavelengths based on the same method at the same frequency. Additionally, we will put even more attention on expanding the working RF frequency to the 5G/6G region or even the terahertz level, developing more RF-generation techniques for comparisons and noise issues at 2 micron wavelengths involving the use of the available mode lock lasers to obtain lower phase noise, applying an optical filter to further suppress the ASE noise induced by the optical amplifier.

**Author Contributions:** Conceptualization, D.J. and N.Y.; methodology, D.J. and N.Y.; software, D.J. and Z.H.; validation, D.J. and Z.H.; formal analysis, D.J. and Z.H.; investigation, D.J. and Z.H.; resources, D.J. and Z.H.; data curation, D.J. and Z.H.; writing—original draft preparation, D.J. and Z.H.; writing—review and editing, D.J. and N.Y.; visualization, D.J and Z.H.; supervision, N.Y.; project administration, F.P. and Y.S.; funding acquisition, F.P., Y.S. and N.Y. All authors have read and agreed to the published version of the manuscript.

**Funding:** This work was supported in part by the National Key Research and Development Program of China (2021YFB2900800, 2019YFB1802901), by the Natural Science Foundation of China (NSFC) (61420106011, 61601277, 61601279), by the Science and Technology Commission of Shanghai Municipality (20511102400, 20ZR1420900), by 111 project (D20031), by STCSM (SKLSFO2019-02), by the Shanghai Pujiang Program (20PJ1404100), and by the National Natural Science Foundation of China (Grant No. 62175143).

**Institutional Review Board Statement:** Not applicable.

**Informed Consent Statement:** Not applicable.

**Data Availability Statement:** Not applicable.

**Acknowledgments:** We also thank Hairun Guo for providing the optical amplifier at 2 micron wavelengths and the technical support of Yongyuan Chu.

**Conflicts of Interest:** The authors declare no conflict of interest. The funders agree to publishing the results.

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
