# Peer review of "Photo-Generation of Tunable Microwave Carriers at 2 µm Wavelengths Using Double Sideband with Carrier Suppression Modulation"

_applsci, doi:10.3390/app12063172_

Round 1

Reviewer 1 Report

Report on manuscript applsci-1586022 Ji:

In the manuscript entitled "Photo-generation of tunable microwave carriers at 2 µm wavelengths using double sideband with carrier suppression modulation" by Ji and coworkers, the authors demonstrated a way to generate 7~12 GHz bandwidth, tunable microwave carriers at 2, 1.55, and 1.31 microns by using the double-sideband carrier suppression (DSB-CS) modulation method both theoretically and experimentally. According to their studies, sideband output power ~ -7.7 dBm was achieved from the Mach-Zehnder modulators for all those three bands with the same indium-rich InGaAs PIN photodetector. Based on their experimental data, they found that the normalized microwave carrier power with respect to the wavelength-dependent squared external quantum efficiency of the photodetector seems close to the simple theoretical values.

After carefully going through the manuscript, I found the presentation must be substantially improved and the English must be polished. There are plenty of grammar issues and typos in the manuscript. All of these make the work hard to read and understand. Although the reported work is highly relevant to applied sciences, yet, before reaching the final decision, I would like to suggest a major revision. In the following, I will specify the major issues appearing in the current manuscript.

(1)  Presentation.  As stated above, the English and presentation of this work must be improved. There are many grammar issues as well as typos. As examples, here I only point out few for your consideration. The sentence from line 40 to line 42 is incomplete. The explanations on Equation (12) in lines 179 -- 184 can be reduced if recall similar notations explained above. Lines 322 -- 324 should be removed from the content.

(2)  Technical issues.  (a) Session 2.1 could be simplified as the theory is well known in frequency modulation. (b) The microwave carrier power P_Eλ (Equation (11)) depends on the initial phase φ. However, through the whole research, it is unclear how the authors handle this phase in their experiments. (c) It would be more instructive if the authors could provide some results on noise issue, especially for the targeted bands after applying the DSB-CS modulations. (d) Although the authors focused on the generated frequency-doubling signals, would the sidebands appearing in Figures 2(a) and 2(b) (see the bumps away from the targeted bands) affect the applications?

(3)  Symbol issues. The symbol VRF in Equation (3) is assumed to be a voltage. Similar issues also go to Vupper and Vlower. No explanation is available for Vπ. In Equation (9), "q" is supposed to be the electron charge "e".

(4)  Reference issues.  There are issues appearing in cited conference proceedings or abstracts. Among them please double check the following references 1, 5, 6, 12, 13, and 21, as part of the information was improper. What is "OptCo" in Ref. 18?

In short, I don't believe the current version meets the high standards of the journal. A substantial revision must be made before a final conclusion could be reached on the reported work.

Reviewer 2 Report

The article is well organized including background, positioning within the literature, theory, and experimental results. However, despite the fact that the general work behind this paper can be grasped, there is a fundamental problem with the English language, which is not clear enough to understand the details. 

One general comment is that the performance at the tested frequency shown seems to be way lower than classical/commercial microwave synthesizers based on standard electronics and 10-MHz reference. Could the authors comment or add a comparison?

Another important missing item is a summary table with the main performance in terms of, e.g., phase noise and dynamic range of the source.

The authors should be better emphasize the novelty and impact of this work. Optical techniques are key at very high frequency, the application at 12 GHz does not look particularly interesting to me.

Also, please add the frequency beyond the "wavelength", at least in the abstract, for a more effective understanding. Sometimes, the "2-um band" wording is used: please add "wavelength" or just mention the frequency.

Reviewer 3 Report

The authors demonstrate tunable microwave carriers at 2 µm wavelengths by applying the DSB-CS modulation method. The comparison is carried out with 1.55 and 1.31 µm. The paper is well written and the quality of this work is good. Some points should be discussed as follows:

1. Is the increasing obtained in the opto-electric power conversion efficiency enough to use the 2 µm wavelengths for 5G/6G applications?

2. The authors employ DSB-CS modulation method to generate the two optical signals, but can the authors using the multi-mode lasers such as MLL at 2 µm to see if we have the same performance, since the MLL is employed a lot in the optical heterodyning systems.

3. The authors mention that 7~12 GHz stable frequency-doubling is measured. In my opinion, the stability in the RF signal after the photodiode is obtained due to the DSB-CS modulation method and not to employing 2 µm wavelengths.

4. In the setup, the optical amplifier is utilized, so how you overcome the ASE noise induced by the amplifier?

5. How long the fiber length is used? It is nice to test the same setup with long fiber length such as 25 km for real applications. Therefore, you can find the impact of the optical phase decorrelation introduced by the optical fiber on the coherent optical spectrum at the MZM output? In this case, the optical phase decorrelation will induce the phase noise on the RF signal. Then, the comparison can be performed between 2 µm wavelengths and the other (1.55 and 1.31 µm) to see if the phase noise from long fiber is higher on the 2 µm wavelengths or not.

Round 2

Reviewer 1 Report

I have carefully gone through the revision and paid particular attention to the response from the authors to the comments in the previous report. I found the authors indeed made notable changes by addressing the issues and comments raised previously. These changes make the reported work more approachable and readable. Moreover, the presentation and English are also improved. The current version now looks satisfactory.  Although English could be further polished, at this stage I would recommend it for publication in Applied Sciences.

Author Response

If we understand the reviewer' comments in the right way, it is not necessary to upload our responses due to that the revision is almost completed according to the opinions.

Reviewer 2 Report

In my opinion, due to English grammar issue and overall style, the manuscript is still not yet enough. 

If the subject of the article is increasing the generation of RF power by means of enhanced opto-electric conversion, then

  • state it clearly in the abstract and introduction; 
  • provide a state-of-the-art of these techniques;
  • add a comparison (table) in terms of RF output power and frequencies by means of opto-electric techniques;
  • it is not clear why THz frequencies and 5G/6G applications are mentioned if the test is only performed at X-band.

In Table 1, please provide a comparison with other commercial equipment.

Reviewer 3 Report

Thanks for the authors. Their responses are clear, and the paper is suitable now for publication.

Author Response

(The authors gave the same response as above.)

Round 3

Reviewer 2 Report

Response 1.

Regarding the English language, the main problem is the verbose style and the lack of focus. It is very hard to understand what you want to mean and to appreciate the work. My suggestion is to perform a complete re-style of the text from an English native speaker. I cannot endorse this paper for publication at the present state.

Response 2.1

OK, then state it clearly without much wording, or the reader will get confused by those many additional info.

Response 2.2

Adding a state-of-the-art does not mean modifying a few words. It means including proper references and discussing why this work add novelty with respect to other techniques.

Response 2.3

My question was related to adding relevant literature comparison and results, not experimental activity. Then, this request has nothing to do with COVID.

Response 2.4.

Please remove any reference to THz or 5G/6G. It is only misleading here, given that there's no experimental result related to these applications.

Response 3.

OK for the table, although you should either provide the performance at the other frequencies or re-shape the layout.